# CRISPR-Cas9-mediated barcode insertion into *Bacillus thuringiensis* for surrogate tracking

Steven A. Higgins,[1] Fadime Kara Murdoch,[1] Jonathon M. Clifton,[1] Jennifer H. Brooks,[1] Keegan L. Fillinger,[1] Jason K. Middleton,[1] Bradley S. Heater[1]

**ABSTRACT** The use of surrogate organisms can enable researchers to safely conduct research on pathogens and in a broader set of conditions. Being able to differentiate between the surrogates used in the experiments and background contamination as well as between different experiments will further improve research efforts. One effective approach is to introduce unique genetic barcodes into the surrogate genome and track their presence using the quantitative polymerase chain reaction (qPCR). In this report, we utilized the CRISPR-Cas9 methodology, which employs a single plasmid and a transformation step to insert five distinct barcodes into *Bacillus thuringiensis*, a well-established surrogate for *Bacillus anthracis* when Risk Group 1 organisms are needed. We subsequently developed qPCR assays for barcode detection and successfully demonstrated the stability of the barcodes within the genome through five cycles of sporulation and germination. Additionally, we conducted whole-genome sequencing on these modified strains and analyzed 187 potential Cas9 off-target sites. We found no correlation between the mutations observed in the engineered strains and the predicted off-target sites, suggesting this genome engineering strategy did not directly result in off-target mutations in the genome. This simple approach has the potential to streamline the creation of barcoded *B. thuringiensis* strains for use in future studies on surrogate genomes.

**IMPORTANCE** The use of *Bacillus anthracis* as a biothreat agent poses significant challenges for public health and national security. *Bacillus anthracis* surrogates, like *Bacillus thuringiensis*, are invaluable tools for safely understanding *Bacillus anthracis* properties without the safety concerns that would arise from using a virulent strain of *Bacillus anthracis*. We report a simple method for barcode insertion into *Bacillus thuringiensis* using the CRISPR-Cas9 methodology and subsequent tracking by quantitative polymerase chain reaction (qPCR). Moreover, whole-genome sequencing data and CRISPR-Cas9 off-target analyses in *Bacillus thuringiensis* suggest that this gene-editing method did not directly cause unwanted mutations in the genome. This study should assist in the facile development of barcoded *Bacillus thuringiensis* surrogate strains, among other biotechnological applications in *Bacillus* species.

**KEYWORDS** CRISPR-Cas9, *Bacillus thuringiensis*, gene insertion, whole-genome sequencing, genetic barcode

*B*acillus anthracis is a Gram-positive, rod-shaped bacterium that is primarily known for its role as the causative agent of anthrax, a highly infectious and potentially lethal disease (1). However, beyond its natural occurrence, *B. anthracis* has also gained notoriety as a biothreat agent due to its ability to be weaponized and used in acts of bioterrorism. The use of *B. anthracis* as a biothreat agent poses significant challenges for public health and national security, and governments and organizations worldwide have

Address correspondence to Bradley S. Heater, heater@battelle.org.

The authors declare no conflict of interest.

See the funding table on p. 11.

implemented stringent measures to enhance preparedness and response capabilities in the event of an intentional release. Surrogates for *B. anthracis* are invaluable tools for understanding *B. anthracis* properties without the safety concerns that would arise from using a virulent strain of *B. anthracis*. Surrogates such as *Bacillus thuringiensis* subsp. *kurstaki* (*Btk*) have been employed extensively due to their similar physiological and biochemical characteristics (2–6). Particularly, dispersion, persistence, and migration of *Btk* spores can be evaluated in nonsterile environments (e.g., outdoors), where *B. anthracis* spores or attenuated strains such as *B. anthracis* Sterne cannot be used due to biosafety requirements and public health concerns. However, a major drawback of experiments utilizing *Bacillus* surrogates in nonsterile environments is the risk of cross-contamination by native *Bacillus* populations.

One way to mitigate cross-contamination is to insert a small, genetically distinct nucleic acid barcode within a neutral region of the genome, which can be tracked specifically by quantitative polymerase chain reaction (qPCR) in nonsterile environments (7). In fact, Buckley et al. describe a set of rules for integrating barcodes into a neutral site of *B. thuringiensis*, (4) while Bernhards et al. designed a tool for orthogonal DNA barcode design (8). However, the insertion method employed in both these reports relies on an inefficient two-plasmid system, where each plasmid is transformed one at a time. As such, a single plasmid system for barcode insertion would significantly reduce the time and effort required to generate barcoded *Bacillus* strains and streamline surrogate tracking efforts.

The CRISPR-Cas9 platform is a gene-editing technology that has been broadly employed to alter the genome of eukaryotes and prokaryotes at specific sites (9, 10). The Cas9 enzyme is directed to the site in the genome by a single-guide RNA (sgRNA), and the Cas9 enzyme makes a double-strand break in the DNA at this site. A homology template containing the desired alteration is presented to the cells to fix the double-strand break by the homology-directed repair (HDR) pathway, which leads to insertion or deletion of the desired gene (10). However, if Cas9 generates a double-stranded break in the DNA at an off-target site, depending on the organism, the break is repaired by the non-homologous end-joining (NHEJ) pathway. The NHEJ system is error-prone and can introduce mutations with deleterious effects on cell survival. However, recent studies on bacteria such as *Bacillus* have suggested that the NHEJ system is only active during sporulation (11), and that it is not activated when CRISPR-Cas9-mediated gene editing was carried out in *B. subtilis* (12).

The CRISPR-Cas9 approach developed by Altenbuchner et al. (13) uses a single-plasmid system for gene editing in *Bacillus subtilis*—a single pJOE plasmid contains all the components required for barcode integration, and gene editing can be performed after a single transformation step. Moreover, this same system has been employed by others for gene editing in *Bacillus cereus*, *Bacillus megaterium,* and *Bacillus anthracis (12, 14–17),* and it has also been used to generate knockout strains of *B. thuringiensis* (18, 19). However, to our knowledge, no CRISPR-Cas9 platform has been employed for gene integration in *B. thuringiensis*. Given the extensive application of the pJOE plasmid for engineering different *Bacillus species*, (13) we surmised it would be an appropriate platform to implement for barcode insertion in *B. thuringiensis*.

Therefore, we applied a CRISPR-Cas9 approach to insert five distinct genetic barcodes into *Bacillus thuringiensis* subsp. *kurstaki (Btk),* an established surrogate for *Bacillus anthracis (8)*. We developed qPCR assays to specifically target and track barcoded *Btk* strains and performed experiments to demonstrate the integrity and stability of the inserted barcodes. We also performed whole-genome sequencing (WGS) on all barcoded strains and wild-type *Btk* to identify mutations and conducted analyses to determine whether those mutations could have been caused by nonspecific Cas9-mediated digestion.

## MATERIALS AND METHODS

### Bacterial strains and plasmids

*Bacillus thuringiensis* subsp. *kurstaki* ATCC 33679 (*Btk*) was purchased from the American Type Culture Collection (ATCC, Manassas, VA, USA). The MAX Efficiency$^T$ DH5α Competent Cells were purchased from Thermo Fisher Scientific, Inc (Waltham, MA, USA). The pJOE8999 plasmid was purchased from The *Bacillus* Genetic Stock Center at The Ohio State University. All strains and plasmids used during this study and their relevant characteristics are listed in (Table S1, Supporting Information). Btk competent cells were prepared by growing an overnight culture at 30°C in 50 mL brain heart infusion (BHI) media. The next day, the cell pellet was washed 3X with 50 mL ice-cold sterile water and resuspended in 1 mL of ice-cold polyethylene glycol (PEG) 6000. Aliquots of 200 µL were snap-frozen on dry ice and stored at −80°C.

### Barcode and sgRNA design

Synthetic barcodes and respective primers were designed using the barcode software (8). The list of barcodes and primers is shown in (Table S2, Supporting Information). The sgRNA sequence used for targeting the Cas9 enzyme to the *Btk* genome was designed with the CRISPOR v 4.98 software (20) and NGG as the protospacer adjacent motif (PAM) sequence. Sequences nearest to the insertion site between positions 4,834,064 and 4,834,065 of NCBI RefSeq accession number NZ_CP010005.1 were selected. The Cas-OFFinder tool (21) using *Bacillus thuringiensis* subsp. *kurstaki* str. HD73 Accession CP004069.1 as the reference genome was used to predict off-site targeting, and sequences were then compared to those of the reference *Btk* genome NZ_CP010005.1 using BLAST. The best 20-bp sgRNA sequence derived from this analysis was 5′-TGAAATG AAATGGTTCAAGT- 3′.

### Construction of pJOE8999 vectors used for barcode insertion

The 20-bp guide sequence was generated by mixing the respective forward and reverse primers and heating at 95°C for 5 minutes and then allowing to cool to RT for annealing. This dsDNA was cloned into the pJOE8999 vector at the *BsaI* site, generating plasmid pJOE8999 sg183. The homology arms used for homologous recombination were synthesized by Integrated DNA Technologies (IDT, Coralville, IA, USA) with *NotI* and *SbfI* restrictions sites that could be used for cloning the barcodes into the pJOE8999 sg183 vector. Approximately 700–850 nucleotides flanking downstream and upstream of the insertion site (between positions 4,834,064 and 4,834,065) in *Btk* ATCC 33679, respectively, were used for the homology region. This sequence was inserted into the pJOE8999 sg183 vector at the *SfiI* sites via NEBuilder HiFi DNA Assembly (New England Biolabs, Ipswich, MA, USA). Barcodes were synthesized by IDT with 25-bp flanking ends to facilitate cloning into the pJOE8999 sg183 vector by NEBuilder HiFi DNA Assembly using *NotI* and *SbfI* restriction sites. Final plasmids used for barcode insertion were named pJOE8999 sg183 *Btk* Bn, where n represents the specific barcode (1, 4, 7, 8, and 12).

### Construction of barcoded strains

Barcode insertion was performed using CRISPR-Cas9 mediated by the pJOE8999 sg183 *Btk* Bn vectors. One microgram of demethylated plasmid DNA was added to the *Btk* competent cells and incubated on ice for 10 minutes. The cell/ DNA mixture was electroporated at 2.5 kV in an Eporator (Eppendorf, Enfield, CT, USA). The cells were recovered in BHI media for 2 hours at 30°C and plated onto Luria Bertani (LB)-agar plates containing 20 µg/mL kanamycin. Single colonies were selected and screened for barcode integration using colony PCR and subsequent restriction digestion with *NotI*. Positive strains were cleared of plasmids by passing 3 x overnight at 42°C on tryptic soy agar (TSA) (Hardy Diagnostics, CA, USA). Generated strains include *Btk* BAR 1, *Btk* BAR 4, *Btk* BAR 7, *Btk* BAR 8, and *Btk* BAR 12.

## Whole-genome sequencing and analysis

The genomic DNA (gDNA) was extracted from cell pellets of the *Btk* wild-type and five barcoded strains using the DNeasy UltraClean Microbial Extraction Kit (Qiagen), or the PacBio Nanobind HMW DNA Extraction protocol using the Nanobind CBB Kit (Pacific Biosciences, Menlo Park, CA, USA). Genomic DNA quality was evaluated using Genomic DNA ScreenTape (Agilent Technologies, Inc., Santa Clara, CA, USA), and DNA was quantified via Qubit 1X dsDNA HS assay (Thermo Fisher Scientific, Inc.).

Illumina libraries were prepared for the *Btk* wild-type and five barcoded strains using the Illumina DNA Prep kit (Illumina, Inc., San Diego, CA, USA) according to the manufacturer's instructions. Pooled Illumina libraries were sequenced on an Illumina NovaSeq 6000 using an SP 2 × 150 flow cell. Nanopore (ONT) sequencing libraries were prepared using the Rapid Barcoding Sequencing kit (Oxford Nanopore Technologies, Oxford, UK) following the manufacturer's protocols. The resulting libraries were pooled in equal volume and adapters added using the Rapid Adapter attachment protocol. ONT sequencing was carried out for 72 hours with fast basecalling and read filtering set to a minimum quality score of 8.

Raw FASTQ files from Illumina and ONT were processed in the software package CLC Genomics Workbench (Qiagen Sciences, Inc.). Raw ONT reads were length-trimmed to remove sequences shorter than 4,000 nt (no quality trimming). Illumina reads were quality-trimmed and a minimum length post-trimming of 140 nt. Assemblies were created using the "*De novo* assembly long reads and polish with short reads" function in CLC Genomics Workbench with default parameters.

The location of barcodes and sequence identity within resequenced *Btk* genomes was confirmed by nucleotide alignment of the barcode sequence to each *Btk* genome using the lastal alignment tool (v1406) (22). Raw Illumina sequence reads were quality-trimmed and filtered using the FastqPuri tool (v1.0.8) with a length threshold of 50 nt and Phred quality cutoff of 30 (23). Reads from each barcoded *Btk* strain (BAR1, BAR4, BAR7, BAR8, and BAR12) were processed using the haploid variant caller Snippy (v4.6.0). Briefly, reads were aligned to both the *de novo*-sequenced wild-type *Btk* strain and a complete reference *Btk* genome [National Center for Biotechnology Information (NCBI) assembly accession no. GCF_000338755.1]. Variants were identified in Snippy using the Freebayes variant caller (24), and variants were finally annotated by SnpEff (25) using provided genome annotations (e.g., using those found in the GenBank file of NCBI Btk Assembly GCF_000338755.1). All tab-delimited variant predictions made by snippy were then combined and imported into RStudio (v2022.07.1 Build 554) (26) and R (v4.2.1) (27) for further analysis.

Predictions of off-target Cas9 sites were generated with the Cas-OFFinder tool (21) using the closed genome of the NCBI *Btk* assembly as a reference. Mutations unique to barcoded strains and the NCBI *Btk* reference genome were compared to off-target Cas9 sites only after removal of shared mutations in our wild-type and barcoded strains compared to the NCBI *Btk* reference. Otherwise, mutation comparisons between barcoded strains and the wild-type *Btk* progenitor strain used in the present study are reported. The Cas-OFFinder parameters were set with mismatches ≤ 4, DNA and RNA bulge ≤1, and Cas9 from *Streptococcus pyogenes* (SpCas9) with PAM of 5′-NGG-3′. All Cas-OFFinder predictions can be found in (Table S3, Supporting Information). Custom Python (v3.9.12) scripts were used to calculate distances between Cas-OFFinder putative off-target CRISPR sites, genetic variants identified in the *Btk* barcoded strains, and identify shared genetic variants in barcoded *Btk* strains and putative PAM sites within six nt downstream of a genetic variant. The Cas9 of the *Streptococcus pyogenes* PAM motif is 5′-NGG-3′ and is included in alignments using the sg183 guide RNA. Global alignments between the 19 nucleotides upstream of the Snippy genetic variant and six nucleotides downstream of the genetic variant and the sg183 guide RNA were performed in Python with the function align.globalxx function from the Biopython (28) pairwise2 module. Conserved domains were identified using the NCBI Batch Web CD Search Tool (29) in the automatic search mode with composition-corrected scoring

and an e-value score cutoff of 0.01. The wild-type *Btk* genome was annotated with Bakta v1.4.2 (30). The tabular Snippy outputs, comprising wild-type and barcoded strains versus NCBI *Btk* genome comparisons and barcoded *Btk* strains versus wild-type *Btk* genome comparisons, are provided in Supplemental Data. For convenience in perusing Snippy comparisons between barcoded and wild-type *Btk* strains, we also provide the Bakta wild-type *Btk* genome annotations in Supplemental Data as well.

## Sporulation/ germination analysis

Wild-type and barcoded *Btk* strains were incubated in BHI broth for 16–18 hours at 30°C. When the $OD_{600}$ was measured to be >1, 2.5 mL of the culture was added to 22.5 mL modified G media (ModG) (1 L contains $(NH_4)_2SO_4$: 2.0 g; $CaCl_2 \cdot 2H_20$: 0.025 g; $CuSO_4 \cdot 5H_20$: 5.0 mg; $FeSO_4 \cdot 7H_20$: 0.5 mg; $MgSO_4 \cdot 7H_20$: 0.2 g; $MnSO_4 \cdot 4H_20$: 0.05 g; yeast extract: 2.0 g; $ZnSO_4 \cdot 7H_20$: 5.0 mg; $K_2HPO_4$: 0.5 g) (31) and incubated at 30°C, for 72 hours to induce sporulation. The spore–cell mixture was washed 2 X in a 30% glycerol solution and then heat-shocked at 60°C for 1 hour to remove vegetative cells. The spore stocks were then enumerated by serially diluting 100 µL in PBS and plating on TSA. One enumeration plate with a single colony was used to inoculate a 5-mL culture of BHI for germination. Five sporulation/ germination passages were performed in triplicate, and following the fifth passage, a colony PCR assay was employed to verify that the barcode was retained in the genome.

## Quantitative PCR analysis

The qPCR analysis was performed on an Applied Biosystems 7500 Fast Real-Time PCR System using the KAPA Universal FAST SYBR Master Mix. Primers used are shown in (Table S2, Supporting Information). Results were analyzed with Applied Biosystems 7500 Fast Software (version 2.3). To check the specificity of various primer sets, a gDNA mix of barcoded strains was prepared, with the exception of the gDNA with which the primers are specific to bind. Calibration curves comprised tenfold serial dilutions of the plasmid DNA with the target barcode sequence and spanned a concentration range from $1.0 \times 10^7$ to $1.0 \times 10^1$ target gene copies $\mu L^{-1}$. All qPCR reactions were carried out in triplicate for each DNA sample with an appropriate set of standards.

## RESULTS AND DISCUSSION

### Barcode insertion into *Btk*

Five genetic barcodes that are orthogonal to *Btk* were designed using the barcode software (8) and then subsequently inserted into the *Btk* genome at a site that was previously shown to be innocuous to the integrity of the cell (4). Barcode insertion was screened by the colony PCR (Fig. 1). Here, the forward primer binds to the genomic DNA upstream of the homology region in the plasmid, and the reverse primer binds in the homology region in the plasmid, which allows the colonies to be screened for barcode insertion prior to plasmid clearing, making the screening process more efficient. Furthermore, a restriction digestion step was incorporated to readily identify differences between wild-type and barcoded strains. As a result, five different strains were generated (*Btk* BAR 1, *Btk* BAR 4, *Btk* BAR 7, *Btk* BAR 8, and *Btk* BAR 12), each with a unique identifier sequence inserted into the genome. Insertion efficiencies of the different synthetic barcodes varied considerably (Table S4, Supporting Information), with the highest efficiency being 60% for B8 and the lowest 5.6% for B4. It is unclear why some barcodes were incorporated more efficiently into *Btk* than others, but additional methodological refinements could improve the efficiency of barcode insertion via this CRISPR-Cas9 approach.

It is crucial that the engineered barcoded stains can be detected by qPCR and that the primers are specific for the respective barcoded strain. The results indicated that each assay designed for B1, B4, B7, B8, and B12 were specific to the target barcode, showed positive amplification with target barcoded gDNA, and did not show nonspecific

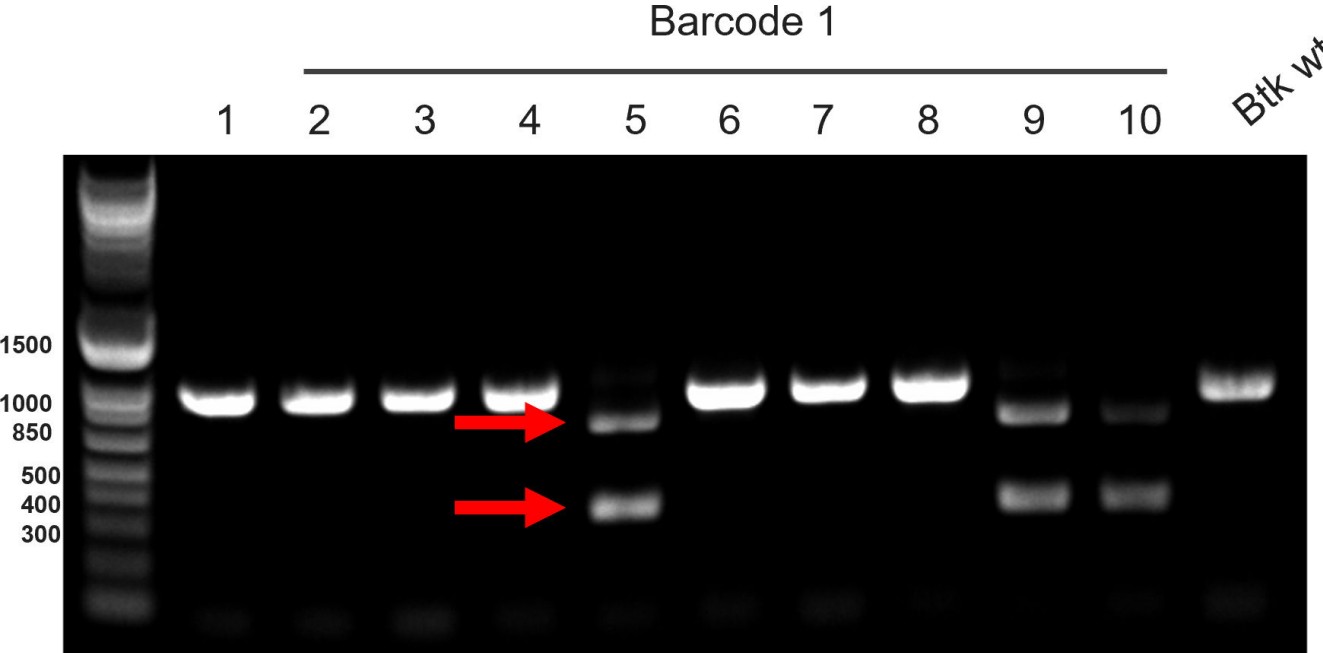

**FIG 1** Colony PCR screen of barcode B1 insertion into *Btk*. Bands digested with the *NotI* restriction enzyme indicate positive barcoded strains and are shown with red arrows.

amplification of other barcoded strains or Btk wild-type. To evaluate the specificity of primers, a gDNA mix was prepared without including the target gDNA. For example, when evaluating the specificity of the B1 primer set, the gDNA mix was prepared via mixing wild-type Btk, B4, B7, B8, and B12 gDNAs and run on the same qPCR plate. All primers pairs were specific to the respective barcodes and showed no cross-reactivity among strains. The compiled qPCR results are shown in (Table 1).

## Genome comparison of barcoded strains to wild-type *Btk*

One potential concern of gene editing is off-site targeting, which could lead to alterations in the genome and result in an unwanted phenotype. Therefore, WGS of the five barcoded strains and *Btk* wild-type was performed. A hybrid approach to WGS was implemented using short-read Illumina sequencing data combined with the long-read Oxford Nanopore technology to maximize coverage and generate complete or high-quality assemblies. Long-read sequencing data allow the elucidation of potential large gene insertions, deletions, or transpositions that are otherwise difficult to assess using only the short-read Illumina approach. The WGS data confirmed the presence of all five barcodes in the respective strains (Table S5, Supporting Information), which corroborates the qPCR data. The barcoded strains were on average 99.97% similar to the sequenced *Btk* wild-type strain, indicating significant alterations in the genomic DNA did not occur after gene editing via CRISPR-Cas9 in this organism.

**TABLE 1** Compiled qPCR results of barcoded strains

| Barcoded strain | Slope | Amplification efficiency (%) | Gene copy abundance (gene copies/mL) (Respective barcode reaction) | Gene copy abundance (gene copies/mL) (Mixture of four other BAR strains and wild-type) | Limit of detection (LOD) (Gene copies/mL) |
|---|---|---|---|---|---|
| *Btk* BAR 1 | −3.271 | 102.174 | $1.41 \times 10^7 \pm 7.86 \times 10^5$ | < LOD | $2.78 \times 10^2$ |
| *Btk* BAR 4 | −3.385 | 97.422 | $1.18 \times 10^7 \pm 5.39 \times 10^5$ | $4.07 \times 10^2 \pm 1.27 \times 10^2$ | $2.78 \times 10^2$ |
| *Btk* BAR 7 | −3.552 | 91.209 | $4.08 \times 10^7 \pm 3.75 \times 10^5$ | < LOD | $2.78 \times 10^3$ |
| *Btk* BAR 8 | −3.107 | 109.843 | $3.40 \times 10^7 \pm 1.60 \times 10^5$ | < LOD | $2.78 \times 10^3$ |
| *Btk* BAR 12 | −3.135 | 108.444 | $1.50 \times 10^7 \pm 1.19 \times 10^6$ | No amplification | $2.78 \times 10^2$ |

To gain further insights into the mutations identified in the engineered strains, the genome of each barcoded strain was compared to the genome of the *Btk* wild-type cells used for barcode insertion (Table S6, Supporting Information). For this particular analysis, only frameshift mutations within a coding sequence were selected since this type of mutation often results in a nonfunctional protein product. The most striking difference found among the barcoded strains was that strain *Btk* BAR 12 had only three frameshift mutations compared to the *de novo*-sequenced *Btk* wild-type (this study), whereas the other four barcoded strains (*Btk* BAR 1, 4, 7, and 8) contained 25–30 frameshift mutations. Interestingly, on average, 95% of the mutations found in these four barcoded strains were shared among three or more barcoded strains (Fig. 2A), and only one unique (not shared by any other barcoded strain) frameshift mutation was identified in *Btk* BAR 4. These data suggest that the majority of these mutations did not occur by random chance due to errors in DNA replication and that *Btk* BAR 1, 4, 7 and 8 may have originated from a genetically distinct *Btk* wild-type strain. Indeed, the first set of barcodes we attempted to integrate into the *Btk* genome were B1, B4, B7, and B8, but *Btk* BAR 12 originated from a different stock of *Btk* ATCC 33679, and the competent cells used for BAR 12 barcode insertion were derived from a different colony.

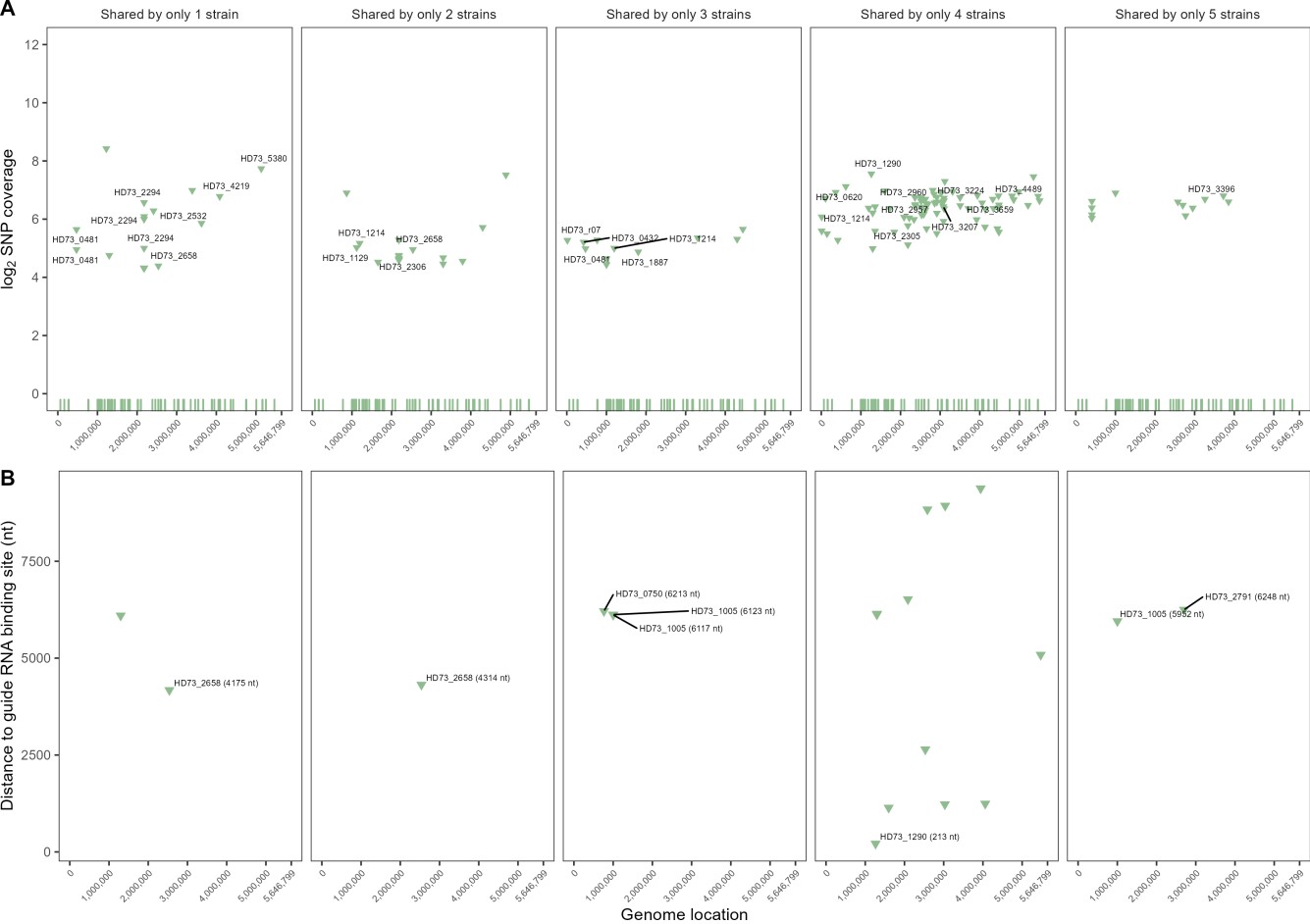

**FIG 2** Colocation of genetic variants and predicted off-target SpCas9 guide RNA-binding sites detected in the genomes of five strains of SpCas9-modified *Bacillus thuringiensis* subsp. *kurstaki* (*Btk*). The log$_2$-transformed coverage and colocation to off-target guide RNAbinding sites of genetic variants shared by up to five SpCas9-modified *Btk* strains (A) and zoomed-in view of the distance between the genetic variants and an off-target guide RNA-binding site. Only variants within 10 kb of a predicted off-target guide RNA-binding site are shown in (B). Green tick marks at the bottom of (A) are the locations of the predicted guide RNA-binding sites. The text annotations prefixed with "HD73" indicate locus tags of genetic variants detected within coding sequences in the Btk genome (NCBI genome accession no. CP004069.1). Values in parentheses next to locus tags in (B) indicate the exact distance of the genetic variant to the nearest off-target guide RNA-binding site.

Still, the large difference in the number of mutations detected between *Btk* BAR 1, 4, 7, and 8 (25–30 mutations) and *Btk* BAR 12 (three mutations) is unclear. The only difference in the CRISPR-Cas9 methods used between the two sets of strains was that the competent cells used for *Btk* BAR 1, 4, 7, and 8 were first frozen at −80°C prior to use, while the competent cells used for *Btk* BAR 12 were transformed fresh. A previous study demonstrated that freeze–thawing can alter genome-scale diversity in *E. coli (*32). As such, it is possible a similar phenomenon occurred here in *Btk,* which could explain the fewer number of frameshift mutations discovered in the *Btk* BAR 12 strain. However, further investigation is necessary to corroborate our findings.

## CRISPR-Cas9 off-target analysis

We also assessed whether the Cas9 enzyme had nonspecifically edited other areas in the genome. We examined mutations found only within Cas9-modified bacterial strains and not detected within their wild-type progenitor and investigated their similarity to the sgRNA and their proximity to predicted off-target Cas9 sites. Although the *de novo* sequence of the wild-type *Btk* strain represents a high-quality draft sequence ($N = 7$ contigs; N50 = 5743666; L50 = 1). For this analysis, we chose to use a closed genome of *Btk* available from the NCBI (accession no. GCF_000338755.1) as a reference for our off-target CRISPR site analyses. We reasoned that we would get more complete detection of off-target CRISPR sites in a closed genome than in a high-quality, fragmented draft. Moreover, the gANI between the wild-type and NCBI *Btk* reference genome is 99.9666%, suggesting the *Btk* NCBI reference genome was suitably identical for this analysis. We removed mutations shared between our sequenced strains (wild-type and barcoded) and the NCBI *Btk* reference genome to home in on mutations specific to the barcoded strains only (which may be due to off-target CRISPR artifacts). This process also minimized the likelihood that mutations detected solely in the barcoded Btk strains arose from genetic drift among our wild-type and the NCBI reference strain. In this analysis, all mutations exclusive to the barcoded *Btk* strains were examined, not just frameshift mutations as reported in the previous analysis. Examination of the distances between 187 putative off-target CRISPR sites detected in the Btk NCBI genome (Cas-OFFinder, Table S3, Supporting Information) and genetic variants shared among the barcoded *Btk* strains revealed the closest putative off-target site to be 213 nt away from a missense variant detected (C->T mutation, Thr424Met) in locus HD73_1290, encoding isocitrate lyase (Fig. 2B). The next nearest variant to a predicted off-target CRISPR site was 1,140 nt away. Therefore, the data do not support a close association between off-target CRISPR-Cas9 site predictions and the genetic variants detected in this study.

The efficacy of off-target CRISPR activity was additionally evaluated by examining whether the six nucleotides downstream of a genetic variant contained a PAM site, which is essential for Cas9 to digest DNA. If a putative PAM site was detected within these six nucleotides upstream of the genetic variant, alignments were performed with the sg183 sgRNA and the 5′-NGG-3′ PAM sequence of the SpCas9 used to create the barcoded *Btk* strains in this study (Fig. 3).

Only five genetic variants possessed PAM sites within six nucleotides downstream of the variant. Of these variants, only four (locations 406410, 1001842, 2859549, and 4815137) demonstrated reasonable pairwise similarity between the sg183 sgRNA, SpCas9 PAM, and the upstream region of the variant (Fig. 3). However, all these sites contain three to four nucleotide mismatches directly adjacent to the PAM site, and complementarity at this four-nt region is critical for SpCas9 digestion (33). Additionally, more than half of the 20-nt potential off-target sites at each site are mismatches, all of which contain at least a span of four or more mismatches in a row. These numerous mismatches significantly increase the Gibbs free energy required for annealing, drastically reducing the ability of the sgRNA to bind to the off-target site. Thus, based on these criteria, it is unlikely that these mutations arose due to off-targeting digestion by SpCas9 and NHEJ-directed repair and the CRISPR-Cas9 methodology applied herein.

```
>CP004069.1_406410-406510,406460,AAGTCAATAATATCAAGGGA,TTTGGC
AAGTC-AA-T-AATAT----CAAGGGATTTGGC
   |  | || | || ||      |||  |  ||||
---T-GAAATGAA-ATGGTTCAA--G--TTGG-
   Score=16

>CP004069.1_1001842-1001942,1001892,CAGGTGAAGGAACAGAAAAG,CCGGAT
CAGGTGAAG--GAACA-GAA----AAGC-CGGAT
   ||||    ||| | |       |||  |||
----TGAA-ATGAA-ATG--GTTCAAG-TCGG--
   Score=15

>CP004069.1_2220679-2220779,2220729,TTTATGTTTTTTATTTTTAT,GGGGGG
TTT-A--TGTTTTTTATTTTTA-TGG-----G-GGG
 |  | ||        |   | |||     | |||
--TGAAATG------A-----AATGGTTCAAGTGGG
   Score=13

>CP004069.1_2859549-2859649,2859599,AATATCTAAAGGGATAGAGG,AAAAGG
--AATATCT-AAAG-GGAT--AGAGGAAA-AGG
  || |  | ||| ||  | | |||      |||
TGAA-A--TGAAA-TGG-TTCA-A-G---TAGG
   Score=16

>CP004069.1_4815137-4815237,4815187,AAATGCAGTATTAAAAAAAG,AACAGG
--AAATGCAGTA-T--TAAAAA--AAGAAC-AGG
  ||||||  |  | |         |||    |||
TGAAATG-A--AATGGT-----TCAAG---TAGG
   Score=15
```

FIG 3  Global alignments between putative sgRNA and PAM sites (top row) and sg 183 (bottom row). The blue underscore identifies the location of the genetic polymorphism detected in three or more barcoded *Btk* strains, whereas the green underscore highlights a putative PAM site within six nucleotides downstream of the genetic variant. The "Score" represents the number of identical characters shared between the two sequences. FASTA-formatted headers are included for reference to the location of the shared genetic variants in the *Btk* genome.

## Barcode stability in the *Btk* genome

Although the WGS data support that the CRISPR-Cas9 approach used for barcode insertion in this study did not result in mutations, it does not account for whether the barcode is stable in the genome. To address this question, five rounds of sporulation and subsequent germination were performed with the five barcoded strains and the wild-type strain. The results demonstrate that all five barcoded strains grew and sporulated similarly to wild-type *Btk* (Fig. 4A). Importantly, the colony PCR assay confirmed that the barcode was retained in the *Btk* chromosome after the five passages, indicating that the barcode insertion is stable in the genome (Fig. 4B).

## Conclusion

We report a simple method for insertion of barcode into *B. thuringiensis* using the CRISPR-Cas9 methodology and subsequent tracking by qPCR. The barcodes can be inserted following a single transformation step and detected by qPCR with high

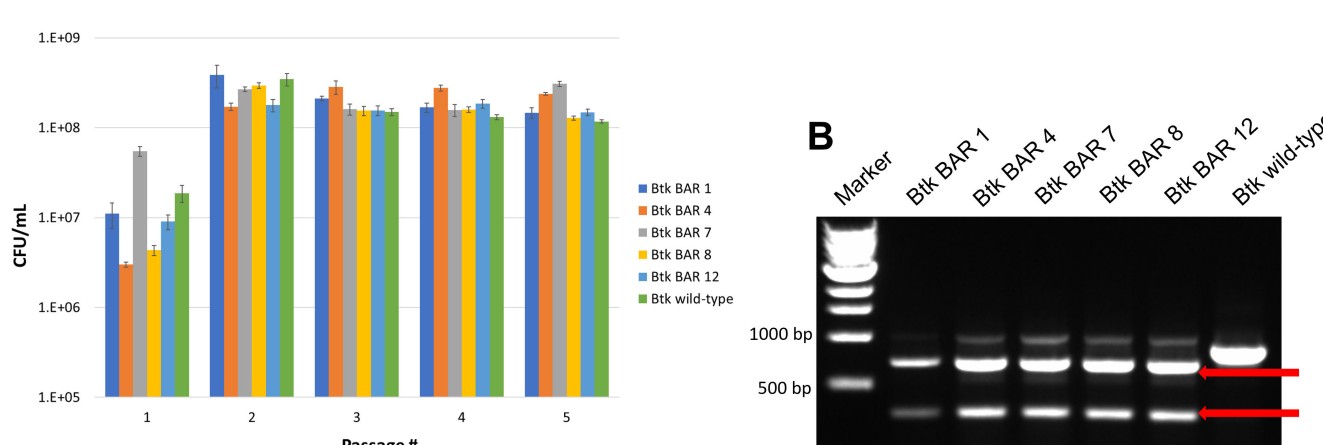

**FIG 4** Sporulation/germination study. (A) Spore counts after each of five passages. (B) Colony PCR assay of barcoded strains after five passages. The red arrows indicate the fragment fingerprint indicative of the barcode after digestion with *Not*I.

sensitivity and specificity. To our knowledge, this is the first time the CRISPR-Cas9 platform has been employed for gene integration into *B. thuringiensis*. The barcodes can remain stable in the genome following multiple rounds of sporulation and germination. Moreover, the WGS data and CRISPR-Cas9 off-target analyses in *Btk* support existing hypotheses that the NHEJ system is inoperable during CRISPR-Cas9-mediated gene editing in *Bacillus* species (11, 12). However, further studies should be performed to more broadly determine the off-target effects of CRISPR in these bacteria, such as employing multiple sgRNAs and insertions at multiple sites within the genome.

## ACKNOWLEDGMENTS

The authors thank the members of the Department of Homeland Security (DHS) Science and Technology Directorate (S&T) Probabilistic Analysis of National Threats, Hazards, and Risks (PANTHR) team for helpful discussions and guidance on study objectives.

This document is based upon work supported by the Defense Technical Information Center (DTIC) under Homeland Defense Technical Area Task (HDTAT) Contract No. FA8075-14-D-0003-FA807518F1414, with funding from DHS S&T through Interagency Agreement (IAA) 70RSAT19KPM0000470001. The views and conclusions contained in this document are those of the authors and should not be interpreted as necessarily representing the official policies, either expressed or implied, of DHS, DTIC, or the U.S. Government. DHS, DTIC, and the U.S. Government do not endorse any products or commercial services mentioned in this report. In no event shall DHS, DTIC, or the U.S. Government have any responsibility or liability for any use, misuse, inability to use, or reliance upon the information contained herein. In addition, no warranty of fitness for a particular purpose, merchantability, accuracy, or adequacy is provided regarding the contents of this document.

The authors would also like to thank the Genomics and Microbiome Core Facility at Rush University for performing whole-genome sequencing.

## AUTHOR AFFILIATION

[1]Applied Science and Technology, Battelle Memorial Institute, Columbus, Ohio, USA

## AUTHOR ORCIDs

Steven A. Higgins  http://orcid.org/0000-0002-5209-5000

Bradley S. Heater  http://orcid.org/0000-0001-8851-4345

## FUNDING

| Funder | Grant(s) | Author(s) |
|---|---|---|
| DOD \| OSD \| Defense Technical Information Center (DTIC) | FA8075-14-D-0003-FA807518F1414 | Steven A. Higgins |
| | | Fadime Kara Murdoch |
| | | Jonathon M. Clifton |
| | | Jennifer H. Brooks |
| | | Keegan L. Fillinger |
| | | Jason K. Middleton |
| | | Bradley S. Heater |
| U.S. Department of Homeland Security (DHS) | 70RSAT19KPM0000470001 | Steven A. Higgins |
| | | Fadime Kara Murdoch |
| | | Jonathon M. Clifton |
| | | Jennifer H. Brooks |
| | | Keegan L. Fillinger |
| | | Jason K. Middleton |
| | | Bradley S. Heater |

## AUTHOR CONTRIBUTIONS

Steven A. Higgins, Data curation, Formal analysis, Investigation, Methodology, Validation, Visualization, Writing – original draft, Writing – review and editing | Fadime Kara Murdoch, Data curation, Formal analysis, Investigation, Methodology, Validation, Writing – original draft, Writing – review and editing | Jonathon M. Clifton, Data curation, Formal analysis, Investigation, Methodology, Writing – review and editing | Jennifer H. Brooks, Data curation, Formal analysis, Investigation, Methodology, Writing – review and editing | Keegan L. Fillinger, Data curation, Formal analysis, Investigation, Methodology, Writing – review and editing | Jason K. Middleton, Conceptualization, Funding acquisition, Project administration, Supervision, Writing – review and editing | Bradley S. Heater, Conceptualization, Data curation, Formal analysis, Funding acquisition, Investigation, Methodology, Supervision, Validation, Visualization, Writing – original draft, Writing – review and editing

## DATA AVAILABILITY

Raw sequencing reads were submitted to the National Center for Biotechnology Information (NCBI) Sequence Read Archive with accessions SRR28699905-SRR28699910 under NCBI BioProject accession no. PRJNA1100612. Draft genome assemblies were submitted to the NCBI GenBank under accession numbers JBEDQN000000000-JBEDQS000000000 under NCBI BioProject accession no. PRJNA1100614. All Snippy analysis outputs for wild-type and barcoded Btk strains and the NCBI Btk reference or the wild-type Btk reference are available in Supplemental_Data.zip.

## ADDITIONAL FILES

The following material is available online.

### Supplemental Material

**Supplemental tables (Spectrum00003-24-s0001.pdf).** Tables S1-S6.

## Open Peer Review

**PEER REVIEW HISTORY (review-history.pdf).** An accounting of the reviewer comments and feedback.

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
