## [Reviewer comments · Microbiology Spectrum]

Microbiology Spectrum

CRISPR-Cas9-mediated barcode insertion into *Bacillus thuringiensis* for surrogate tracking

Steven Higgins, Fadime Kara-Murdoch, Jonathon Clifton, Jennifer Brooks, Keegan Fillingner, Jason Middleton, and Bradley Heater

Corresponding Author(s): Bradley Heater, Battelle Memorial Institute

Review Timeline:

Submission Date:	January 9, 2024
Editorial Decision:	February 24, 2024
Revision Received:	April 22, 2024
Accepted:	May 7, 2024

Editor: Silvia Cardona

Reviewer(s): Disclosure of reviewer identity is with reference to reviewer comments included in decision letter(s). The following individuals involved in review of your submission have agreed to reveal their identity: Emilia Ghelardi (Reviewer #3)

Transaction Report:

DOI: <https://doi.org/10.1128/spectrum.00003-24>

Re: Spectrum00003-24 (CRISPR-Cas9-mediated barcode insertion into *Bacillus thuringiensis* for surrogate tracking)

Dear Dr. Bradley Scott Heater:

Thank you for submitting your manuscript to Microbiology Spectrum. Your article has been reviewed by three experts in the field. Reviewers found your work technically sound and have provided comments that, if addressed, will improve the quality of the manuscript. Specifically, I would like you to ensure all genome sequencing data is deposited in public repositories and previous work on barcoded microbial systems and CRISPR-Cas9 in *B. thuringiensis* are put into the context of this work and properly cited in the introduction. In addition, reconsider the statement that *B. thuringiensis* can be used for *Bacillus anthracis* research in the context of current knowledge in the field and their genomic differences.

Reviewers' recommendations are provided below.

Revision Guidelines

Sincerely,
Silvia Cardona
Editor
Microbiology Spectrum

Reviewer #1 (Comments for the Author):

The authors convincingly demonstrate that Altenbuchner's pJOE system for single-step genome editing is effective for generating barcoded strains of a *Bacillus thuringiensis* surrogate for *B. anthracis*. The methodology generates specific barcodes that can readily be detected by qPCR, and it does not simultaneously generate unwanted off-site mutations. The manuscript is clearly written. I have a couple of concerns that the Methods section is not sufficiently detailed to allow the experiments to be repeated. I also wonder if including parental strain genome sequences in the study would be a good idea. Further, my own preference is that the five genome sequences obtained during the study be deposited in a publicly accessible database. Beyond that, there are only a few minor points about taxonomic nomenclature and use of proper names to correct.

*** Necessary elaborations of the Methods section:

(1) line 143: What method was used to prepare the Btk competent cells? What other parameters were used for the electroporation besides the voltage setting?

(2) lines 202-210: What is the recipe or reference for ModG? Please also elaborate on the full name and source of TSA.

*** Preferable, but not mandatory, additional experiment:

(3) lines 265-287: It would have been best practice to determine the genome sequence of the parental strain(s) used, since slight heterogeneity among deposits of the same strain in different repositories is to be expected. Is it too late for to obtain those data? The authors' hypothesis that differing experimental conditions, such as freeze-thawing, accounted for the sequence diversity is theoretically possible, genetic drift of the parent strains seems much more likely to me. It would be a simple hypothesis to test.

*** Minor edits

(4) line 58: Technically, it is not necessary to introduce the abbreviated genus parenthetically here. You can simply write the full genus and species name when the taxon is first used, then use the abbreviated form for all subsequent uses in the text. But this is a very minor point, of course.

(5) line 89: The term Bacilli should not be employed as a plural of the genus name. Better would be "Bacillus species" or something similar. The word "Bacilli" should just be a reserved for describing "cells with a rod-like shape."

(6) line 93: *B. thuringiensis* does not have recognized subspecies. Designations like "kurstaki" reflect distinct flagellar serotypes, not subspecies status. More correct would be "*Bacillus thuringiensis* serovar. kurstaki" here.

(7) line 181: "GenBank" is more correct than "genbank."

(8) lines 188, 195: Since "Python" is a proper names for software, it should begin with an uppercase letter.

Reviewer #2 (Comments for the Author):

Barcode ms

1. L.27-what is evidence that B t is "a well established surrogate"? Are the genomes so extensively identical to rule out differences in physiology? BT and Ba probably grow in sufficiently different environments that they use different carbon sources, etc. Why not use the Ba Sterne strain as a surrogate?

2. Line 142- any special conditions to make Bt competent?

3. Line 203-what is ModG? Explain or reference.

4. Fig. 4B no obvious red arrow

5. Single CrispR plasmid an improvement over mating used in ref 2 especially eliminating deletions and non-site insertions. Stability also of value. But concerns over use of Bt especially when non-toxic Sterne strain available.

Reviewer #3 (Comments for the Author):

General comments

Barcoded microbes can be helpful in tracking microorganism in the environment and many studies were done using the CRISPR-Cas9 system to this aim.

In this study, the authors applied the CRIPR-Cas9 system for barcode insertion in *Bacillus thuringiensis*. The idea was already proposed without the use of the CRISPR-Cas9 system in this organism and the authors should better define how their study gives fundamental novelties to the field.

I found complete lack of an introduction to barcoded microbial systems (i.e . Qian et al., Science 368, 1135-1140 (2020) and all the applications of CRISPR-Cas9 in *B. thuringiensis*. These are examples of the papers I found: Modulation of Cas9 level for efficient CRISPR-Cas9-mediated chromosomal and plasmid gene deletion in *Bacillus thuringiensis*. Biotechnol Lett. 2020; Detection and Tracking of a Novel Genetically Tagged Biological Simulant in the Environment. Applied and Environmental Microbiology 78.

In addition, the authors state that *B. thuringiensis* can be considered an invaluable tool for understanding *Bacillus anthracis* properties. I think this is not true. The organisms are pretty different and the crucial pleiotropic regulator PlcR is present in Bt and inactive in Ba.

As regards Ba, many studies were produced using the CRISPR-Cas system (i.e. Highly Efficient Genome Engineering in *Bacillus anthracis* and *Bacillus cereus* Using the CRISPR/Cas9 System. Front Microbiol. 2019; Efficient Genome Editing by a Miniature CRISPR-AsCas12f1 Nuclease in *Bacillus anthracis*. Front Bioeng Biotechnol. 2022; CRISPR-Cas12a assisted recombinase based strand invading isothermal amplification platform designed for targeted detection of *Bacillus anthracis* Sterne, International Journal of Biological Macromolecules, 2024; A CRISPR/Cas12a-based DNzyme visualization system for rapid, non-electrically dependent detection of *Bacillus anthracis*, Emerging Microbes & Infections, 2024).

Finally, insufficient description of the obtained results (as well as discussion) is present in the manuscript.

Additional comments

Lines 227-228: Can the site be described?

Lines 238-240: Please add discussion on this aspect. Is there any other study describing the same effect?

Sporulation/germination are not sufficient for evaluating defects of the barcoded strains. The authors at least should show growth dynamics, biochemical profile, motility behavior.

Genome sequences should be submitted to a public sequence repository.

Reviewer #1 (Comments for the Author):

The authors convincingly demonstrate that Altenbuchner's pJOE system for single-step genome editing is effective for generating barcoded strains of a Bacillus thuringiensis surrogate for B. anthracis. The methodology generates specific barcodes that can readily be detected by qPCR, and it does not simultaneously generate unwanted off-site mutations. The manuscript is clearly written. I have a couple of concerns that the Methods section is not sufficiently detailed to allow the experiments to be repeated. I also wonder if including parental strain genome sequences in the study would be a good idea. Further, my own preference is that the five genome sequences obtained during the study be deposited in a publicly accessible database. Beyond that, there are only a few minor points about taxonomic nomenclature and use of proper names to correct.

Thank you for your critical evaluation of the manuscript – we really appreciated your feedback and input and feel that it has greatly improved the quality of our manuscript.

Based on your feedback regarding data submission to a public repository, we have now created a “Data Availability” section and have included the following sentence in the main manuscript: “Raw sequencing reads were submitted to the National Center for Biotechnology Information (NCBI) Sequence Read Archive with accessions SRR28699905-SRR28699910 under NCBI BioProject accession PRJNA1100612. Draft genome assemblies were submitted to NCBI GenBank under accessions XXXXX-XXXXX. All Snippy analysis outputs for wild-type and barcoded Btk strains and the NCBI Btk reference or the wild-type Btk reference are available in Supplemental_Data.zip.”

The raw sequence read data was accepted shortly after submission (NCBI accession numbers SRR28699905-SRR28699910 under BioProject accession PRJNA1100612), but the WGS data submission is still in a “Queued” status. We are currently in contact with NCBI staff to understand our position in the queue and how long it will take for us to generate NCBI accession numbers for the six submitted genomes. Communication has been slow from NCBI staff, but we will gladly add the NCBI accessions for the WGS data to the manuscript as soon as we hear back from NCBI staff or our data reach “Processed” status and accession numbers are generated on the NCBI Submission Portal.

***** Necessary elaborations of the Methods section:**

(1) line 143: What method was used to prepare the Btk competent cells? What other parameters were used for the electroporation besides the voltage setting?

Thank you for identifying this gap in our methods reporting. We're happy to provide that information and have added the following text to the revised manuscript on **lines 114-117**.

Btk competent cells were prepared by growing an overnight culture at 30°C in 50 mL Brain Heart Infusion (BHI) media. The next day the cell pellet was washed 3X with 50 mL ice-cold

sterile water and resuspended in 1 mL of ice-cold polyethylene glycol (PEG) 6000. Aliquots of 200 µl were snap frozen on dry ice and stored at -80°C.

Great question! Unlike some Electroporators, it is not possible to adjust parameters such as resistance and capacitance on the Eporator (Eppendorf). Therefore, if someone wanted to repeat this electroporation protocol all they would need is the voltage (kV) and equipment (Eporator), and both were listed in the methods (**lines 152-153**). We also read through the Eporator manual to determine if any other technical information was given so that this method could be performed on another electroporator besides an Eporator, but we did not find any such details.

(2) lines 202-210: What is the recipe or reference for ModG? Please also elaborate on the full name and source of TSA.

Modified G media (ModG) (1 L contains $(\text{NH}_4)_2\text{SO}_4$ - 2.0 g; $\text{CaCl}_2 \cdot 2\text{H}_2\text{O}$ - 0.025 g; $\text{CuSO}_4 \cdot 5\text{H}_2\text{O}$ - 5.0 mg; $\text{FeSO}_4 \cdot 7\text{H}_2\text{O}$ - 0.5 mg; $\text{MgSO}_4 \cdot 7\text{H}_2\text{O}$ - 0.2 g; $\text{MnSO}_4 \cdot 4\text{H}_2\text{O}$ - 0.05 g; Yeast Extract - 2.0 g; $\text{ZnSO}_4 \cdot 7\text{H}_2\text{O}$ - 5.0 mg; K_2HPO_4 - 0.5 g). This was included on **lines 224-227** and a reference given.

TSA was defined previously (**line 157**), but the source was added (**line 157**).

***** Preferable, but not mandatory, additional experiment:**

(3) lines 265-287: It would have been best practice to determine the genome sequence of the parental strain(s) used, since slight heterogeneity among deposits of the same strain in different repositories is to be expected. Is it too late for to obtain those data? The authors' hypothesis that differing experimental conditions, such as freeze-thawing, accounted for the sequence diversity is theoretically possible, genetic drift of the parent strains seems much more likely to me. It would be a simple hypothesis to test.

We apologize for any confusion, but we did indeed sequence the genome of the wild-type Btk strain used for barcode insertion in the present study along with the five barcoded strains. We have now updated language in the methods on **lines 189-195, 198-202, and 214-219** to clearly indicate that the wild-type and the five barcoded strains were sequenced. We've also clearly identified the use of both the high-quality draft wild-type Btk genome sequence we generated ($N = 7$ contigs, $N50 = 5,743,666$, $L50 = 1$) and the complete NCBI reference genome (Accession GCF_000338755.1). For off-target CRISPR site analyses, we excluded from analysis shared mutations detected in our the wild-type and barcoded strains and the NCBI Btk reference genome to isolate mutations unique to the barcoded strains (and potentially due to off-target CRISPR artifacts). We reasoned that using the closed NCBI Btk genome to detect off-target CRISPR sites was preferable to our incomplete, albeit high-quality, reference genome. Furthermore, locus tags in the NCBI Btk reference genome may be more familiar to the Btk community for comparative analysis purposes than invalid locus tags present in our resequenced Btk strain assembly.

Since we have excluded genetic variation due solely to differences in the wild-type and NCBI reference genomes, we feel that the effect of genetic drift is minimized in our analysis. Moreover, the frameshift mutation analysis in which the freeze-thaw hypothesis was proposed

was performed between the barcoded strains and our wild-type Btk strain and is not influenced by drift between the NCBI Btk strain and our wild-type strain.

We have added information to the results and discussion section on **lines 325-339** which we hope clears up any confusion on this issue. We sincerely appreciate the thoughtful questions by the reviewer and believe that these discussions have greatly clarified the analyses reported in the manuscript.

***** Minor edits**

(4) line 58: Technically, it is not necessary to introduce the abbreviated genus parenthetically here. You can simply write the full genus and species name when the taxon is first used, then use the abbreviated form for all subsequent uses in the text. But this is a very minor point, of course.

Thank you for catching that – we agree and have removed the parenthetical reference on **lines 51 and 58**.

(5) line 89: The term Bacilli should not be employed as a plural of the genus name. Better would be "Bacillus species" or something similar. The word "Bacilli" should just be a reserved for describing "cells with a rod-like shape."

We completely agree and totally understand the confusion here. Thank you for the revision suggestion. We have now edited the sentence to say "*Bacillus species*" on **line 94**.

(6) line 93: *B. thuringiensis* does not have recognized subspecies. Designations like "kurstaki" reflect distinct flagellar serotypes, not subspecies status. More correct would be "*Bacillus thuringiensis servor. kurstaki*" here.

While we do agree that kurstaki may represent an invalidly published subspecies of *B. thuringiensis*, we politely decline to change the nomenclature used in the present study. We believe that listing it as a serovar in the present manuscript will unnecessarily confuse readers interested in reproducing this work or working with strains of *B. thuringiensis* subsp. *kurstaki* overall. For example, this name is listed in multiple culture collections as a subspecies, not serovar, and in the List of Prokaryotic names with Standing in Nomenclature (LPSN, <https://lpsn.dsmz.de/>). The LPSN does list it as invalidly published but does not provide more accurate nomenclature to refer to. Furthermore, the invalid name *B. thuringiensis* subsp. *kurstaki* is listed (incorrectly, as we both agree) in countless publications, and although incorrect, represents a historical oversight which has not been rectified in microbiology literature and is outside the scope of this work to correct. This issue should be brought to the attention of the International Committee on Systematics of Prokaryotes (ICSP) and proper actions taken to rectify this issue. In the process of drafting this response, we attempted to identify comparative genomics analyses or other comparative analyses that may have tried to rectify this nomenclatural issue via the ICSP or proposals in the International Journal of Systematic and Evolutionary Microbiology (IJSEM), but none could be identified. However, articles were found in which *B. thuringiensis* subsp. *kurstaki* is correctly listed as a serovar rather than the invalid subspecies. Unfortunately, these nomenclatural issues are widespread in microbiological

literature. With recent additional nomenclatural proposals made by both the Genome Taxonomy DataBase (GTDB) and the National Center for Biotechnology Information (NCBI), we may be dealing with lots of nomenclatural issues for the foreseeable future.

(7) line 181: "GenBank" is more correct than "genbank."

We agree and have now edited genbank to GenBank on **line 193**. Thank you for catching this error.

(8) lines 188, 195: Since "Python" is a proper names for software, it should begin with an uppercase letter.

Agreed. We have now edited python to Python on **lines 205 and 211**.

Reviewer #2 (Comments for the Author):

Barcode ms

1. L.27-what is evidence that B t is "a well established surrogate"? Are the genomes so extensively identical to rule out differences in physiology? BT and Ba probably grow in sufficiently different environments that they use different carbon sources, etc. Why not use the Ba Sterne strain as a surrogate?

Thank you for raising these great points and we hope we can better explain our rationale for using Btk as a surrogate for Ba. Being prepared for a potential attack involves understanding how spores disperse, persist, and migrate when released into the environment. However, due to public health concerns and biosafety regulations, it's not possible to conduct outdoor tests with live *B. anthracis* spores, or even attenuated strains such as *B. anthracis* Sterne since it is a Risk Group 2 organism. Therefore, closely related Risk Group 1 species are used as substitutes. Several studies have employed Btk spores as a Risk Group 1 surrogate for *B. anthracis* spores due to its similar physiological and biochemical characteristics. Supporting text and 5 references were included in Abstract (**line 28**) the introduction (**lines 60-64**) to support the use of Btk as a surrogate for *B. anthracis*.

2. Line 142- any special conditions to make Bt competent?

We apologize for this oversight in our methods reporting and have now added the following text to **lines 114-117** of the revised manuscript: Btk competent cells were prepared by growing an overnight culture at 30°C in 50 mL Brain Heart Infusion (BHI) media. The next day the cell pellet was washed 3X with 50 mL ice-cold sterile water and resuspended in 1 mL of ice-cold polyethylene glycol (PEG) 6000. Aliquots of 200 µl were snap frozen on dry ice and stored at -80°C.

3. Line 203-what is ModG? Explain or reference.

Thank you again for catching this oversight. Modified G media (ModG) (1 L contains (NH₄)₂SO₄ - 2.0 g; CaCl₂ · 2H₂O - 0.025 g; CuSO₄ · 5H₂O - 5.0 mg; FeSO₄ · 7H₂O - 0.5 mg; MgSO₄ · 7H₂O - 0.2 g; MnSO₄ · 4H₂O - 0.05 g; Yeast Extract - 2.0 g; ZnSO₄ · 7H₂O - 5.0 mg; K₂HPO₄ - 0.5 g). We

have now included this text on **lines 224-227** and have provided a reference.

4. Fig. 4B no obvious red arrow

We apologize for any confusion caused by this issue. We have now increased the size of the red arrows in Figure 4 to make them readily visible in the gel.

5. Single CrispR plasmid an improvement over mating used in ref 2 especially eliminating deletions and non-site insertions. Stability also of value. But concerns over use of Bt especially when non-toxic Sterne strain available.

Thank you for these comments regarding the CRISPR techniques employed in our manuscript. We believe the final comment to be related to concerns already raised in question 1. Although attenuated *B. anthracis* Sterne is a surrogate, it is classified as a Risk Group 2 organism and cannot be safely released into the environment to study spore dispersal, persistence, and resuscitation. Therefore, Btk is used as a surrogate and we have provided additional references listing Btk as a surrogate for this purpose and additional text to address this (**lines 60-64**). The reviewer's concerns are valid, but safety is the primary concern when working with potentially dangerous microorganisms. Although Btk may not be the most closely related available surrogate, it represents a low-risk alternative that enables testing in realistic scenarios which can better inform risk analysis.

Reviewer #3 (Comments for the Author):

General comments

Barcoded microbes can be helpful in tracking microorganism in the environment and many studies were done using the CRISPR-Cas9 system to this aim.

In this study, the authors applied the CRIPR-Cas9 system for barcode insertion in Bacillus thuringiensis. The idea was already proposed without the use of the CRISPR-Cas9 system in this organism and the authors should better define how their study gives fundamental novelties to the field.

Thank you for your comments and we agree that we could better define to the Microbiology Spectrum readership of the novelty our manuscript provides. We also agree that the use of a barcode for simulant tracking is not the novelty of this manuscript since it was already documented in Buckley et. al. (2012). We believe the novelty of this work is the use of the CRISPR-Cas9 platform to generate barcoded strains, and to our knowledge this is the first report that describes CRISPR-Cas9 mediated gene insertion into *Bacillus thuringiensis*. We have now included a statement in the Conclusion (**lines 375-377**) to better highlight the novelty of this work.

I found complete lack of an introduction to barcoded microbial systems (i.e . Qian et al., Science 368, 1135-1140 (2020) and all the applications of CRISPR-Cas9 in B. thuringiensis. These are examples of the papers I found: Modulation of Cas9 level for efficient CRISPR-Cas9-mediated chromosomal and plasmid gene deletion in Bacillus thuringiensis. Biotechnol Lett. 2020; Detection and Tracking of a Novel Genetically

Tagged Biological Simulant in the Environment. Applied and Environmental Microbiology 78.

We apologize for any inadvertent omissions of literature cited in the introduction, and we thank the reviewer for providing additional references. We included the Qian et al., Science 368, 1135-1140 (2020) reference in the manuscript (**line 69**). The following sentence (**lines 69-71**) also describes the use of barcoded *Btk* strains. We also added additional references in the first paragraph of the introduction (for more information on this see the next query), including the additional reference suggested (Detection and Tracking of a Novel Genetically Tagged Biological Simulant in the Environment. Applied and Environmental Microbiology 78). The use of CRISPR-Cas9 in *Bacillus thuringiensis* was previously described on **lines 91-92**, but we also included the “Modulation of Cas9 level for efficient CRISPR-Cas9-mediated chromosomal and plasmid gene deletion in *Bacillus thuringiensis*. Biotechnol Lett. 2020” reference here as well. We hope the additional text we have added, and the inclusion of these new references has improved the background on barcoded microbial systems reported in the introduction.

In addition, the authors state that B. thuringiensis can be considered an invaluable tool for understanding Bacillus anthracis properties. I think this is not true. The organisms are pretty different and the crucial pleiotropic regulator PlcR is present in Bt and inactive in Ba.

Being prepared for a potential attack involves understanding how spores disperse, persist, and migrate when released into the environment. However, due to public health concerns, it's not possible to conduct outdoor tests with live *B. anthracis* spores, or even attenuated strains such as *B. anthracis* Sterne since it is a Risk Group 2 organism. Therefore, closely related Risk Group 1 species are used as substitutes. Several studies have employed Btk as a Risk Group 1 surrogate for *B. anthracis* due to its similar physiological and biochemical characteristics. Supporting text and 5 references were included in the introduction (**lines 60-64**) to support the use of Btk as a surrogate for *B. anthracis*.

As regards Ba, many studies were produced using the CRISPR-Cas system (i.e. Highly Efficient Genome Engineering in Bacillus anthracis and Bacillus cereus Using the CRISPR/Cas9 System. Front Microbiol. 2019; Efficient Genome Editing by a Miniature CRISPR-AsCas12f1 Nuclease in Bacillus anthracis. Front Bioeng Biotechnol. 2022; CRISPR-Cas12a assisted recombinase based strand invading isothermal amplification platform designed for targeted detection of Bacillus anthracis Sterne, International Journal of Biological Macromolecules, 2024; A CRISPR/Cas12a-based DNzyme visualization system for rapid, non-electrically dependent detection of Bacillus anthracis, Emerging Microbes & Infections, 2024).

Thank you for providing these additional references for the CRISPR-Cas work performed in *B. anthracis*. In addition to (*Highly Efficient Genome Engineering in Bacillus anthracis and Bacillus cereus Using the CRISPR/Cas9 System. Front Microbiol. 2019*) that we had previously, we also added other provided references into the manuscript (**Lines 90-92**).

Finally, insufficient description of the obtained results (as well as discussion) is present in the manuscript.

We hope that the significant changes made to the Results and Discussion section, particularly to the sequencing analysis, has improved upon the description of the obtain results.

Additional comments

Lines 227-228: Can the site be described?

The site is between positions 4,834,064 and 4,834,065 of NCBI RefSeq accession number NZ_CP010005.1, which is described in the Methods section (**line 140**).

Lines 238-240: Please add discussion on this aspect. Is there any other study describing the same effect?

Since this is the first report showing barcode into *Bacillus thuringiensis* using CRISPR-Cas9, to our knowledge there is not any precedent in the literature describing the same effect in this organism. One potential explanation for the differences in insertion efficiencies of the varying barcodes could be due to differences in secondary structure. The secondary structure of the guide sequence has been known to affect editing efficiencies (Corsi, G. I.; Qu, K.; Alkan, F.; Pan, X.; Luo, Y.; Gorodkin, J. *CRISPR/Cas9 gRNA activity depends on free energy changes and on the target PAM context. Nat Commun* 2022, 13 (1), 3006. DOI: 10.1038/s41467-022-30515-0), so it is possible that secondary structure of the homology template could affect insertion. However, since this is highly speculative, we decided not to include this in the manuscript.

Sporulation/germination are not sufficient for evaluating defects of the barcoded strains. The authors at least should show growth dynamics, biochemical profile, motility behavior.

The intention of doing the sporulation/ germination study was not to evaluate defects of the barcoded strains (this is one reason for running whole-genome sequencing), but rather to ensure that the incorporated barcode was retained in the genome following multiple rounds of sporulation/ germination. Some text in this section was modified to better illustrate the significance of this experiment (**lines 366-367**).

Genome sequences should be submitted to a public sequence repository.

Thank you for this feedback and we apologize for neglecting this step and not including it sooner. We have now added a Data Availability section to the main manuscript text and have added text (**lines 247-254**), "Raw sequencing reads were submitted to the National Center for Biotechnology Information (NCBI) Sequence Read Archive with accessions SRR28699905-SRR28699910 under NCBI BioProject accession PRJNA1100612. Draft genome assemblies were submitted to NCBI GenBank under accessions XXXXX-XXXXX. All Snippy analysis outputs for wild-type and barcoded Btk strains and the NCBI Btk reference or the wild-type Btk reference are available in Supplemental_Data.zip."

The raw sequence read data was accepted shortly after submission (NCBI accession numbers SRR28699905-SRR28699910 under BioProject accession PRJNA1100612),

but the WGS data submission is still in a “Queued” status. We are currently in contact with NCBI staff to understand our position in the queue and how long it will take for us to generate NCBI accession numbers for the six submitted genomes. Communication has been slow from NCBI staff, but we will gladly add the NCBI accessions for the WGS data to the manuscript as soon as we hear back from NCBI staff or our data reach “Processed” status and accession numbers are generated on the NCBI Submission Portal.

Re: Spectrum00003-24R1 (CRISPR-Cas9-mediated barcode insertion into *Bacillus thuringiensis* for surrogate tracking)

Dear Dr. Bradley Scott Heater:

Your manuscript has been accepted, and I am forwarding it to the ASM production staff for publication. Your paper will first be checked to make sure all elements meet the technical requirements. Please note that there are some inconsistencies in the use of the terms subspecies vs. serovar (legend of figure 2, GenBank: CP004069.1). Please ensure the inconsistencies are corrected in the final version.

ASM staff will contact you if anything needs to be revised before copyediting and production can begin. Otherwise, you will be notified when your proofs are ready to be viewed.

Please note that the term subspecies vs. serovar has some inconsistencies (legend of figure 2, GenBank: CP004069.1). Please ensure the inconsistencies are corrected in the final version.

Sincerely,
Silvia Cardona
Editor
Microbiology Spectrum

Reviewer #1 (Comments for the Author):

In this revision, the authors have substantially improved the manuscript and have incorporated the reviewers' main suggestions.

Reviewer #2 (Comments for the Author):

1. What is meant by "genetically distinct Bt wild type"? Is every colony from a streaked plate different?

Is it possible to repeat insertion with another unfrozen colony?

2. Not convinced of suitability of Btk as a surrogate. There is nothing in the references cited to say otherwise especially given "gaps" cited by Greenberg et al. Might be best to try and get Sterne strain reclassified especially since it has been used extensively in numerous publications.

3. Methods and analysis for undesired mutations seems valid but need a more convincing surrogate.